# 3D Convolutional Neural Network-Based Denoising of Low-Count Whole-Body ^18^F-Fluorodeoxyglucose and ^89^Zr-Rituximab PET Scans

**DOI:** 10.3390/diagnostics12030596

**Published:** 2022-02-25

**Authors:** Bart M. de Vries, Sandeep S. V. Golla, Gerben J. C. Zwezerijnen, Otto S. Hoekstra, Yvonne W. S. Jauw, Marc C. Huisman, Guus A. M. S. van Dongen, Willemien C. Menke-van der Houven van Oordt, Josée J. M. Zijlstra-Baalbergen, Liesbet Mesotten, Ronald Boellaard, Maqsood Yaqub

**Affiliations:** 1Cancer Center Amsterdam, Department of Radiology and Nuclear Medicine, Vrije Universiteit Amsterdam, Amsterdam UMC, De Boelelaan 1117, 1081 HV Amsterdam, The Netherlands; s.golla@amsterdamumc.nl (S.S.V.G.); g.zwezerijnen@amsterdamumc.nl (G.J.C.Z.); os.hoekstra@amsterdamumc.nl (O.S.H.); yws.jauw@amsterdamumc.nl (Y.W.S.J.); m.huisman@amsterdamumc.nl (M.C.H.); gams.vandongen@amsterdamumc.nl (G.A.M.S.v.D.); j.zijlstra@amsterdamumc.nl (J.J.M.Z.-B.); r.boellaard@amsterdamumc.nl (R.B.); maqsood.yaqub@amsterdamumc.nl (M.Y.); 2Cancer Center Amsterdam, Department of Hematology, Vrije Universiteit Amsterdam, Amsterdam UMC, De Boelelaan 1117, 1081 HV Amsterdam, The Netherlands; 3Cancer Center Amsterdam, Department of Medical Oncology, Vrije Universiteit Amsterdam, Amsterdam UMC, De Boelelaan 1117, 1081 HV Amsterdam, The Netherlands; c.menke@amsterdamumc.nl; 4Faculty of Medicine and Life Sciences, Hasselt University, Agoralaan Building D, B-3590 Diepenbeek, Belgium; liesbet.mesotten@zol.be; 5Department of Nuclear Medicine, Ziekenhuis Oost Limburg, Schiepse Bos 6, B-3600 Genk, Belgium

**Keywords:** low-count, CNN, denoising, ^18^F-FDG, ^89^Zr-antibody

## Abstract

Acquisition time and injected activity of ^18^F-fluorodeoxyglucose (^18^F-FDG) PET should ideally be reduced. However, this decreases the signal-to-noise ratio (SNR), which impairs the diagnostic value of these PET scans. In addition, ^89^Zr-antibody PET is known to have a low SNR. To improve the diagnostic value of these scans, a Convolutional Neural Network (CNN) denoising method is proposed. The aim of this study was therefore to develop CNNs to increase SNR for low-count ^18^F-FDG and ^89^Zr-antibody PET. Super-low-count, low-count and full-count ^18^F-FDG PET scans from 60 primary lung cancer patients and full-count ^89^Zr-rituximab PET scans from five patients with non-Hodgkin lymphoma were acquired. CNNs were built to capture the features and to denoise the PET scans. Additionally, Gaussian smoothing (GS) and Bilateral filtering (BF) were evaluated. The performance of the denoising approaches was assessed based on the tumour recovery coefficient (TRC), coefficient of variance (COV; level of noise), and a qualitative assessment by two nuclear medicine physicians. The CNNs had a higher TRC and comparable or lower COV to GS and BF and was also the preferred method of the two observers for both ^18^F-FDG and ^89^Zr-rituximab PET. The CNNs improved the SNR of low-count ^18^F-FDG and ^89^Zr-rituximab PET, with almost similar or better clinical performance than the full-count PET, respectively. Additionally, the CNNs showed better performance than GS and BF.

## 1. Introduction

^18^F-fluorodeoxyglucose (^18^F-FDG) positron emission tomography (PET) is essential in staging of a broad spectrum of malignancies [1,2,3]. Currently, a whole-body ^18^F-FDG PET scan is acquired using a scan duration of 2 min per bed position and an injected activity of 3.7 MBq/kg. A shorter scan duration per bed position could ideally decrease the total scan duration, and therefore, minimize movement artefacts and increase patient comfort and throughput. A reduction of injected activity would decrease the radiation burden for the patient, and therefore makes it possible to perform more frequent ^18^F-FDG PET scans per patient for restaging and therapy-response assessments, in case of scanning children and/or for non-oncological cases. However, a shorter scan duration and lower injected activity would result in a lower signal-to-noise ratio (SNR). Poor scan quality due to low-count (LC) is also observed for ^89^Zr-antibody PET scans, which are obtained after relatively low injected activity imposed by the radiation burden of ^89^Zr [4]. Therefore, denoising LC whole-body ^18^F-FDG and ^89^Zr-antibody PET scans is of interest for improving image quality.

Traditionally, Gaussian smoothing (GS) has been used to denoise PET images [5]. However, GS reduces the spatial resolution of the images, and therefore, could impair detectability and quantification of small (tumour) lesions [6]. Bilateral filtering (BF) exhibits superior properties in comparison to the more commonly used GS for noise reduction in PET [7]. BF reduces the noise of PET scans, while preserving spatial information (e.g., edges). However, BF parameters are difficult to optimize in a generic way because both an optimized intensity and spatial parameter need to be determined, which depend on both the tracer and site of interest. Therefore, another adaptive/data-driven denoising method with high accuracy is warranted.

Convolutional Neural Networks (CNN) are a specialized type of Neural Networks that use convolution to extract features from the PET scan. This is done by convolution filters which assign importance/weights to (learnable) features present in the PET scan. It can therefore learn and detect features such as PET intensities, edges, shapes, etc. Therefore, CNNs are highly beneficial in various medical image processing/segmentation tasks [8,9,10,11]. A CNN-based deep-learning algorithm may also be superior in denoising tasks since it can learn non-linear latent/hidden (not observable by humans) features (which you want to preserve) from LC PET scans and increase the SNR [12,13,14,15,16,17,18]. Therefore, denoising LC whole-body ^18^F-FDG and ^89^Zr-antibody PET scans using a CNN may be performed to improve the SNR and thus their diagnostic value. In previous studies [12,13,14,17,19] a successful application of CNNs for improving ^18^F-FDG PET scans has been presented. However, these studies were performed in a small (oncology) patient cohort, were based on unsupervised deep learning networks and on improving full-count (FC) ^18^F-FDG PET scans or a longer scan duration per bed position.

Therefore, the aim of this study was to develop, train, and extensively evaluate the performance of CNNs to denoise LC whole-body ^18^F-FDG and ^89^Zr-antibody PET scans. A secondary aim was to compare the diagnostic value of the CNNs to that of GS and BF.

## 2. Materials and Methods

### 2.1. Participants

We included PET scans of 60 patients with stage I–IV non-small-cell lung carcinoma (NSCLC) (40 patients from Limburg PET-Center Hasselt Belgium (LPC) [20], and 20 patients from Amsterdam UMC, location VUmc), of which five patients with diffuse large B cell lymphoma (DLBCL) non-Hodgkin lymphoma (Amsterdam UMC, location VUmc) [21] (Table 1). The study at LPC was registered at clinical trials.gov, NCT02024113. The data from the patients with lung cancer at Amsterdam UMC were retrospectively obtained from medical records, with a waiver for informed consent from the Medical Ethics Review Committee of Amsterdam UMC, location VUmc. This study was registered as IRB2018.029. The patients with non-Hodgkin lymphoma were included as part of studies performed by Jauw et al. These patients provided written informed consent, and the studies were approved by the Medical Ethics Review Committee of Amsterdam UMC, location VUmc. This study was registered at Dutch Trial Register http://www.trialregister.nl (accessed on 19 January 2022), NTR3392.

### 2.2. Data Acquisition

Whole-body ^18^F-FDG PET scans in LPC were acquired with a Gemini Big Bore TF PET/CT scanner, and in Amsterdam UMC with an Ingenuity TF PET/CT and Vereos Digital PET/CT scanners (Philips Medical Systems, Best, The Netherlands). ^89^Zr-rituximab PET scans (patients with non-Hodgkin lymphoma) were acquired with an Ingenuity TF PET/CT. For ^18^F-FDG PET scans, 60 min after 259.4 ± 43.8 MBq tracer injection, a low-dose computed tomography (LDCT) scan was performed for attenuation correction and anatomical localisation, and subsequently a 20 min static (exact time depends on patient length) whole-body ^18^F-FDG PET scan was acquired (2 min per bed position). Six days after the injection of 73.7 ± 0.3 MBq ^89^Zr-rituximab, an LDCT scan was obtained, directly followed by a 60 min static whole-body PET scan (5 min per bed position). Corrections for decay, dead time, normalization (detector sensitivities), attenuation, random coincidences and scatter were applied.

Amsterdam UMC ^18^F-FDG PET data were reconstructed with a 10 s (super-low-count (SLC), 92% scan time reduction), 30 s (low-count (LC), 75% scan time reduction) and 2 min (full-count (FC)) scan duration per bed-position. The (S)LC PET scans were reconstructed using multiple time points/delays, which was later used for data augmentation during training. These scans were reconstructed using the blob-basis function ordered-subsets time of flight (BLOB-OS-TF) for the Ingenuity TF PET/CT scanner and the novel ordered subset expectation maximization (OSEM 3i15s, 1i6r-PSF, 4 mm FWHM GAUSS, OSEM 3i15s, 3 mm FWHM GAUSS) for the Vereos Digital PET/CT scanner. The ^89^Zr-rituximab scans, and the ^18^F-FDG PET data from LPC were reconstructed with a FC 5 min and 2 min scan duration per bed position only using BLOB-OS-TF, respectively.

The ^18^F-FDG and ^89^Zr-rituximab PET scans from Amsterdam UMC were reconstructed according to current European Association of Nuclear Medicine Research Ltd. (Vienna, Austria) EARL1 standards and settings associated with EARL accreditation [22], respectively. Matrix and voxels sizes were 144 × 144 and 4 mm in all directions, respectively. ^18^F-FDG PET scans from LPC were reconstructed according to EARL1 standards, with matrix and voxel sizes of 169 × 169 and 4 mm, respectively.

### 2.3. Image Processing

For each FC whole-body ^18^F-FDG PET scan from LPC, SLC PET, scans were simulated using the SiMulAtion and ReconsTruction (SMART)-PET package [23]. Simulation-reconstruction settings were chosen so that the simulated noisy ^18^F-FDG PET scans from LPC showed an almost similar coefficient of variation as the SLC-reconstructed ^18^F-FDG PET scans from Amsterdam UMC. The simulated PET images were used to initially train the CNN, while parts of the actual reconstructed images were used for further fine-training of the CNN, details are explained later.

### 2.4. Model Architecture

A supervised U-Net based [11] 3D-CNN (Figure 1 and Appendix B) was used to denoise the (S)LC 18F-FDG PET scans while maintaining their diagnostic value. However, instead of the max-pooling layer that is traditionally used, in this study the down sampling layers consisted of convolution layers with a stride of two [24]. Although the convolution layer compresses the feature image just as is the case for a max-pooling layer, it does not exclude voxels by only looking at the maximum values. It therefore, not only reduces computation time (although less than max-pooling), but most importantly increases the model its ability to learn [24]. Additionally, in contrast to conventional CNNs, a kernel size of 6 × 6 × 6 instead of 3 × 3 × 3 was applied to learn inter-slice morphological features [25].

### 2.5. Model Performance

#### 2.5.1. Quantitative Performance

The simulated SLC whole-body ^18^F-FDG PET scans from LPC were used to pre-train a 3D-CNN. Next, the reconstructed SLC and LC ^18^F-FDG PET data from Amsterdam UMC were used for fine-training (transfer-learning) the pre-trained model, which generated two additional models (SLC-CNN and LC-CNN) that are tailored to manage low or super low count/quality images. These two models were subsequently used for further evaluation. Training of the CNN model on the simulated noisy LPC ^18^F-FDG data was performed to avoid overfitting due to the small dataset. Noise characteristics of ^89^Zr-rituximab and LC ^18^F-FDG PET scans were almost similar. However, we used the SLC-CNN to denoise the ^89^Zr-rituximab PET scans instead of the LC-CNN, because of the higher level of noise reduction.

The ^18^F-FDG PET data from LPC was split into a training (80%, *n* = 32) and a validation (20%, *n* = 8) set. Thereafter, for further refinement, validation and testing, the two CNN models, ^18^F-FDG and ^89^Zr-rituximab data from Amsterdam UMC, were used. During this training, the data were split into a training (32%, *n* = 8), a validation (8%, *n* = 2) and an independent test (60%, *n* = 15) set. The training and validation set from Amsterdam UMC consisted of only ^18^F-FDG PET scans from the Ingenuity TF PET/CT scanner. The test set, however, consisted both of ^18^F-FDG PET scans from the Ingenuity TF PET/CT scanner, the Vereos Digital PET/CT scanner, and ^89^Zr-rituximab PET scans from the Ingenuity TF PET/CT scanner. PET data augmentation was applied during each training epoch (train-data only) by randomly sampling the different (time points/delays) (S)LC ^18^F-FDG PET scans for each patient during training. In other words, instead of traditional augmentation (shifts, zoom, translation, rotation, etc.), in each training epoch, minor differences in noise characteristics were present.

To compare the performance of the CNNs with other denoising methods, the (S)LC test PET scans were also denoised using traditional GS (^18^F-FDG) and more advanced BF [17] (^18^F-FDG and ^89^Zr-rituximab) denoising methods (Table 2). A Mann–Whitney U test (*p* < 0.05) was used to compare tumour recovery coefficients (TRC) and levels of noise in the images after applying the denoising methods.

We calculated TRC for the ^18^F-FDG and ^89^Zr-rituximab PET scans (test data) using Equation (1). TRC was computed for the test data post-processed with a 3D-CNN, GS, or BF denoising method and compared this to the FC data. PET uptake features from the tumour volumes were extracted (*U_X_*, *X* = average, maximum and 3Dpeak) for both the denoised $\left(U_{X}\;denoised\right)$ as the *FC* $\left(U_{X}\;FC\right)$ PET scans, using the in-house built and open-access ACCURATE tool (quAntitative onCology moleCUlaR Analyses SuiTE) [26,27]. From the ^18^F-FDG scans, only the primary lung tumour was extracted using a 50% SUV3Dpeak isocontour (Table 1 and Figure A1). For the ^89^Zr-rituximab PET scans, tumours were extracted using manual delineation (Table 1 and Figure A1). For patients with non-Hodgkin lymphoma with more than three tumours, bootstrapping was applied to randomly choose three tumours for analysis.
(1)$$\mathrm{TRC}=\frac{U_{X}\;denoised}{U_{X}\;FC}$$

The level of noise was presented as the coefficient of variance (*COV*; Equation (2)). Four spherical volume of interest (VOIs) were drawn in the liver (because the liver showed homogeneous tracer uptake in this cohort, and therefore, could be used to reliably assess the level of noise). Average standard deviation $\left(\bar{\sigma \;liver}\right)$ and average uptake $\left(\bar{U_{avg}\;liver}\right)\;$ were extracted using these four VOIs.
(2)$$COV=\frac{\bar{\sigma \;liver}}{\bar{U_{avg}\;liver}}$$

#### 2.5.2. Qualitative Performance

For a qualitative assessment of the denoising methods (CNN and BF), the images after denoising were independently evaluated by two experienced nuclear medicine physicians (BZ and OH). GS was not included in this assessment due to a mostly significant (*p* < 0.05) lower quantitative performance in comparison to the CNNs and BF. The questionnaire was drafted to assess the reliability and effectiveness of the denoising methods. The assessment was blinded, i.e., the scans presented to the physicians were a random combination (without labels) of the FC, SLC, LC (with and without denoising) PET scans per patient. The ^18^F-FDG and ^89^Zr-rituximab PET scans were scored per patient (1–5: low to high) based on the level of noise, tumour detectability, overall scan quality, clinical acceptability (yes/no), and overall best performance (1st/2nd/3rd/4th/(5th)). A Mann–Whitney U test (*p* < 0.05) was used to compare the performance of the denoising methods.

## 3. Results

### 3.1. Quantitative Assessment

#### 3.1.1. ^18^F-FDG

The BF and the proposed CNN (SLC- and LC-CNN) denoised PET scans have an overall higher TRC and more similar COV to the FC PET scans than GS (Figure 2 and Table A1). In contrast with BF, the SLC-CNN denoised PET scans showed a higher average uptake TRC, a higher 3Dpeak uptake TRC, but a lower maximum uptake TRC. With regard to the LC scans, the LC-CNN denoised PET scans showed a trend (0.05 < *p* < 0.1) of a higher TRC than the BF denoised PET scans for the average uptake, and 3Dpeak uptake. Additionally, the LC-CNN showed a higher but not significant maximum uptake TRC than the BF. In addition, the LC-CNN denoised PET scans showed a similar COV as the FC PET scans. The SLC-CNN had the second closest COV to the FC PET scans.

#### 3.1.2. ^89^Zr-Rituximab

The SLC-CNN denoised PET scans showed a predominant trend of a TRC higher than the 3 mm and 4 mm BF (Figure 3 and Table A2). The SLC-CNN even showed a significantly (*p* < 0.05) higher average uptake TRC than the 3 mm and 4 mm BF. The 3 mm and 4 mm BF presented a comparable COV as the SLC-CNN. The 2 mm BF had a significantly (*p* < 0.05) higher COV than the 3 mm and 4 mm BF and the SLC-CNN.

### 3.2. Qualitative Assessment

For the ^18^F-FDG scans, the observers found lower levels of noise, better tumour detectability, better overall scan quality and higher clinical acceptability for all the CNN models in comparison to the BF denoising methods (Figure 4 and Figure A2, Table 3), with the only exception being SLC-CNN in terms of noise levels and tumour detectability.

For the ^89^Zr-rituximab scans, the observers found a comparable level of noise, but similar/better tumour detectability, better overall scan quality and higher clinical acceptability for the SLC-CNN in comparison to the BF denoising methods (Figure 5 and Table 4).

## 4. Discussion

CNN models to denoise (S)LC ^18^F-FDG and ^89^Zr-rituximab PET scans were trained and extensively evaluated. Overall, the CNN models performed better than the conventional GS and the more advanced BF denoising methods for both the ^18^F-FDG and ^89^Zr-rituximab PET scans. As such, the CNN models show promise for reducing the acquisition time and injected activity of ^18^F-FDG PET scans and increasing the image quality of ^89^Zr-rituximab PET scans.

In this study, we trained noise-specific CNN models, to address the difference in noise levels seen for different scan acquisition times and injected activity in ^18^F-FDG PET scans. However, in case of PET tracers such as ^89^Zr-antibody PET, training a noise specific CNN model was not feasible. Due to the dose limits of ^89^Zr, the overall image quality was impaired (low SNR), and therefore, no high quality ^89^Zr-antibody PET images were available for training a CNN. So, the only possible solution was to directly apply the SLC-CNN (trained using SLC ^18^F-FDG PET scans) to the ^89^Zr-rituximab PET scans and test its performance. The main advantage of this approach is that this validation is the ultimate way of externally testing the CNN on data that are obtained with a different tracer. Although the SLC-CNN is not trained on ^89^Zr-rituximab PET scans, it obtained a higher TRC than the 3 mm and 4 mm BF denoising methods (Figure 3 and Table A2). 

With regard to ^18^F-FDG PET scans with a low injected tracer activity, such as scans with shorter scan duration, a lower SNR will be observed, which impairs both the quantitative and qualitative value of these scans. The CNNs could therefore also be useful to maintain a good image quality when reducing injected ^18^F-FDG activity in whole-body ^18^F-FDG PET studies, and therefore, reduce radiation burden for the patient, but maintain diagnostic value. However, further assessment is necessary to evaluate the performance of CNNs when used for a reduction in the injected activity for whole-body ^18^F-FDG PET acquisitions.

The qualitative assessment also showed that the proposed CNNs were preferred over BF. However, the CNN denoised (S)LC ^18^F-FDG PET scans did show an overall lower qualitative performance than the FC ^18^F-FDG PET scans. Yet, the LC-CNN denoised LC ^18^F-FDG PET scans obtained a similar clinical acceptability score as the FC ^18^F-FDG PET scans (Table 3), while for ^89^Zr-rituximab PET scans, the SLC-CNN increased the overall image quality of the FC ^89^Zr-rituximab PET scans (Table 4). The observers preferred the SLC-CNN denoised ^89^Zr-rituximab PET scans over both the BF denoised and FC ^89^Zr-rituximab PET scans. This can be explained by a higher ratio between tumour signal and background signal present in the SLC-CNN denoised ^89^Zr-rituximab PET scans (Figure 5). This indicates that the SLC-CNN shows promise for establishing an optimal denoising setup for ^89^Zr-antibody PET scans.

In this study, several strategies to prevent overfitting were applied. First, data augmentation was applied by randomly sampling the different (S)LC ^18^F-FDG PET scans for each patient. By using traditional augmentation, interpolation may be different between the training data ((S)LC) and training labels (FC), and therefore, this was not applied in this study. Another method by which overfitting was reduced is by using the symmetric connections in the U-Net based 3D-CNN [28]. As shown in previous studies [12,13], training a model using a small dataset could result in overfitting. Since acquiring sufficient real (S)LC ^18^F-FDG PET data were not feasible, SLC ^18^F-FDG PET data were generated using the already available LPC data. SLC ^18^F-FDG PET data from LPC were simulated using SMART, which facilitated the development of a pre-trained model familiar with morphological features. This resulted in a shorter learning time, lower probability of overfitting, and a more accurate and robust model. Even though pre-training of the model was only performed on SLC ^18^F-FDG PET data from LPC, the fine-trained LC-CNN showed a higher performance than a LC-CNN without a pre-trained model.

The main limitation of this study is the size of the patient cohort. Small-sized tumours in the (S)LC PET scans are more prone to being underestimated by the proposed CNNs. This is because the training data were devoid of small tumours. Therefore, it could be that the model specifies this signal as noise rather than a tumour-specific signal [29]. However, the proposed CNNs showed better correspondence with the FC PET scans than GS and BF. So, although small tumours were present in a small number in the training data, by using the proposed CNNs, more quantitative information was retained in comparison to GS and BF. Even though the differences in performance between the proposed CNNs and BF were small, contrary to BF, a CNN still has the ability to learn and improve by incorporating more patients. Thus, further evaluation in a larger and more heterogeneous cohort could further improve CNNs performances. However, although the proposed method showed promising results for denoising low-count ^18^F-FDG PET scans, obtaining a similar quantitative and qualitative value as the FC ^18^F-FDG PET scans may not be fully feasible and we therefore foresee that the main applications of the CNNs are denoising and improving image quality of ^89^Zr-antibody PET studies.

## 5. Conclusions

The 3D-CNNs used in this study to denoise (S)LC whole body ^18^F-FDG and ^89^Zr-rituximab PET scans were constructed and tested. The CNN denoised (S)LC ^18^F-FDG and ^89^Zr-rituximab PET scans showed almost similar or better clinical performance than the FC scans, respectively. Therefore, the proposed CNNs show promise for reducing PET scan duration or lowering injected activity of whole-body ^18^F-FDG PET scans but are particularly useful to increase the quantitative and qualitative image quality of ^89^Zr-rituximab PET scans.

## Figures and Tables

**Figure 1 diagnostics-12-00596-f001:**
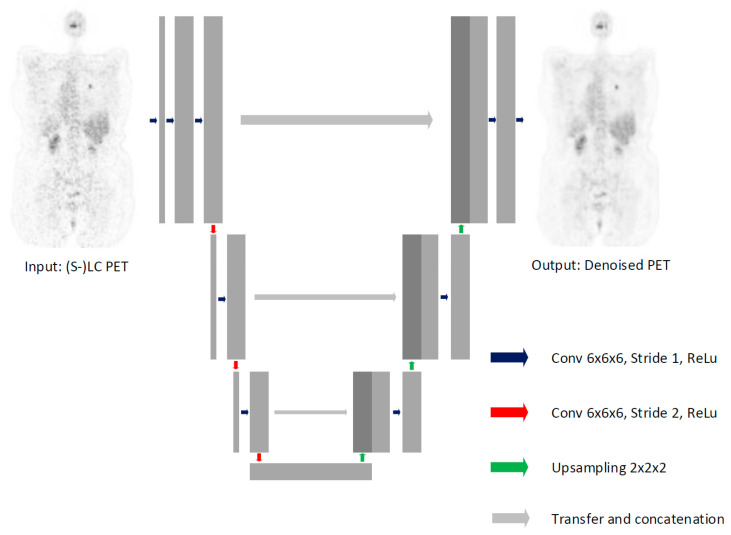
Architecture of the U-net shaped 3D-CNN used in the study. It consists of an encoding and decoding path, which are connected with concatenation layers at each resolution block.

**Figure 2 diagnostics-12-00596-f002:**
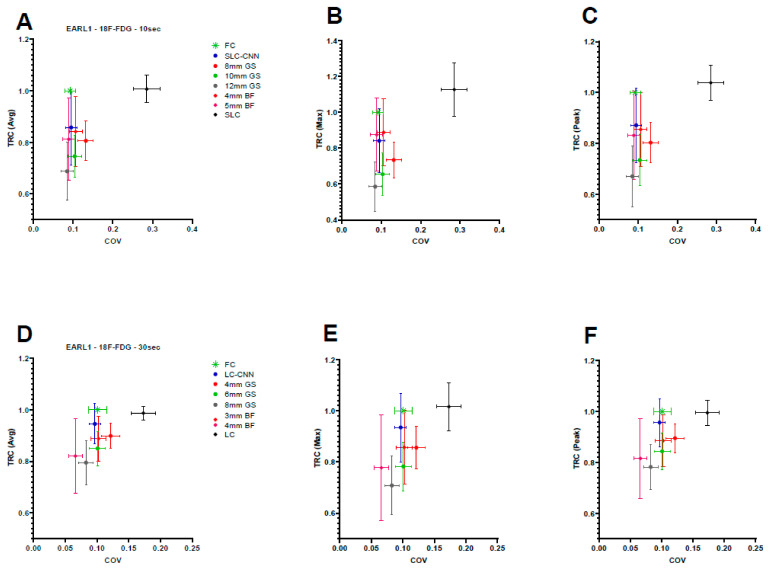
The performance of the denoising methods for low-count ^18^F-FDG PET. The (**A**) average, (**B**) maximum, (**C**) 3Dpeak TRC of the SLC-CNN, GS (8 mm, 10 mm and 12 mm) and BF (4 mm and 5 mm) denoising methods of the SLC ^18^F-FDG PET from the Ingenuity TF PET/CT and the Vereos Digital PET/CT scanner. The (**D**) average, (**E**) maximum, (**F**) 3Dpeak TRC of the LC-CNN, GS (4 mm, 6 mm and 8 mm) and BF (3 mm and 4 mm) denoising methods of the LC ^18^F-FDG PET from the Ingenuity TF PET/CT and the Vereos Digital PET/CT scanner.

**Figure 3 diagnostics-12-00596-f003:**
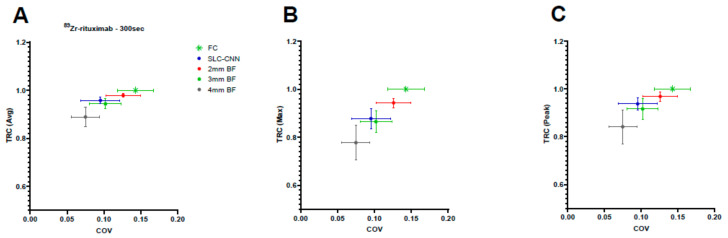
The performance of the denoising methods for low-count ^89^Zr-rituximab PET. The (**A**) average, (**B**) maximum, (**C**) 3Dpeak TRC of the SLC-CNN and BF (2 mm, 3 mm and 4 mm) denoising methods of the FC ^89^Zr-rituximab PET from the Ingenuity TF PET/CT scanner.

**Figure 4 diagnostics-12-00596-f004:**
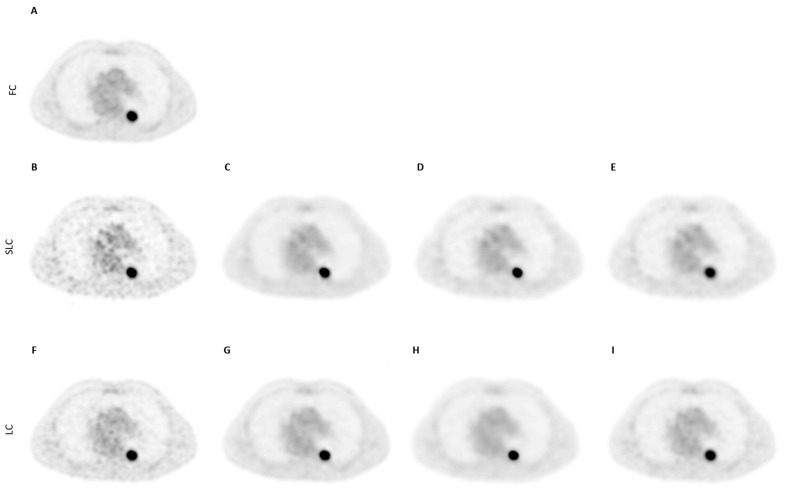
Illustration of a (**A**) FC, (**B**) SLC, (C) SLC-CNN, (**D**) BF 4 mm, (**E**) GS 10 mm denoised ^18^F-FDG PET scan (axial orientation) from the Ingenuity TF PET/CT scanner. Illustration of a (**F**) LC, (**G**) LC-CNN, (**H**) BF 4 mm, (**I**) GS 6 mm denoised ^18^F-FDG PET scan (axial orientation) from the Ingenuity TF PET/CT scanner.

**Figure 5 diagnostics-12-00596-f005:**
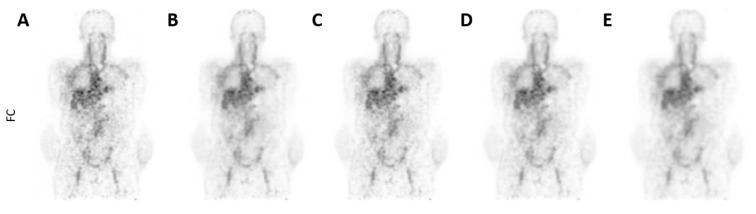
Illustration of a (**A**) FC, (**B**) SLC-CNN, (**C**) BF 3 mm, (**D**) BF 4 mm, (**E**) BF 5 mm denoised ^89^Zr-rituximab PET scan (coronal orientation) from the Ingenuity TF PET/CT scanner.

**Table 1 diagnostics-12-00596-t001:** The patients characteristics included in this study from Amsterdam UMC.

	^18^F-FDG (*n* = 20) NSCLC–Amsterdam UMC	^18^F-FDG (*n* = 40) NSCLC–LPC	^89^Zr-Rituximab (*n* = 5)Non-Hodgkin Lymphoma
Male/female (*n*)	11/9	25/15	3/2
Injected dose (MBq)	259.4 ± 43.8	298.2 ± 49.4	73.7 ± 0.3
Tumour volume (cubic centimeter (cc))–Test data	29.2 ± 54.3	-	25.6 ± 51.3

**Table 2 diagnostics-12-00596-t002:** Gaussian smoothing (GS) and Bilateral Filtering (BF) settings evaluated for the denoising of the low-count whole-body PET scans.

	GS (FWHM)	BF (FWHM; SUV)
^18^F-FDG PET		
SLC	8 mm, 10 mm and 12 mm	4 mm and 5 mm; SUV2.5
LC	4 mm, 6 mm and 8 mm	3 mm and 4 mm; SUV2.5
^89^Zr-rituximab PET	6 mm, 8 mm and 10 mm	2 mm, 3 mm and 4 mm; SUV2.5

**Table 3 diagnostics-12-00596-t003:** Scores provided by the Nuclear Medicine Physicians as part of qualitative assessment of the denoised ^18^F-FDG PET scans. The best performing method is indicated in bold based on the average score of both physicians.

	SLC			LC			FC
Metrics [1–5: Low–High]	SLC-CNN	BF-4 mm	BF-5 mm	LC-CNN	BF-3 mm	BF-4 mm	
Level of noise	**3.0–3.2**	4.0–4.0 *	3.6–4.0 *	**1.8–2.0**	2.6–2.0 **	2.2–3.8 *	1–1 *
Tumour detectability	2.0–3.0	**2.2–3.0**	2.0–3.0	**4.0–4.0**	3.6–4.0	2.8–3.0 *	5–5 *
Overall scan quality	**2.4–2.6**	1.6–2.0 *	1.8–2.0 *	**4.4–4.0**	3.8–4.0	2.8–2.2 *	5–5 *
Clinically acceptable? [%]	**0–80**	0–0 *	0–0 *	**100–100**	80–100	20–0 *	100–100
Best scan (1/2/3/4)	**2–2**	3–3	4–4	**2–3**	**3–2**	4–4	1–1

* significant (*p* < 0.05) higher/lower than (S)LC-CNN. ** trend (*p* < 0.1).

**Table 4 diagnostics-12-00596-t004:** Qualitative assessment of the ^89^Zr-rituximab PET scans. Scores were given for the PET scans with (SLC-CNN and BF) and without (FC) denoising by both Nuclear Medicine Physicians. In bold the best performing method (or scan) is indicated based on the average score of both physicians.

Metrics [1–5: Low–High]	SLC-CNN	BF-2 mm	BF-3 mm	BF-4 mm	FC
Level of noise	2.4–2.6	3.8–4.6 *	2.8–2.6	**1.4–1.2 ***	4.6–4.8 *
Tumour detectability	3.4–3.8	4.4–4.0 *	2.4–2.4 *	1.4–1.4 *	**4.6–4.2 ***
Overall scan quality	**3.8–3.8**	3.6–3.8	3.4–3.0 *	2.0–1.4 *	3.4–3.0 *
Clinical acceptable? [%]	**100–100**	**100–100**	80–80	**0–0 ***	80–80
Best scan (1/2/3/4/5)	**1–1**	3–2	2–4	4–5	3–3

* significant (*p* < 0.05) higher/lower than SLC-CNN.

## Data Availability

Data can make available on reasonable request.

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
