# Peer review of "3D Convolutional Neural Network-Based Denoising of Low-Count Whole-Body 18F-Fluorodeoxyglucose and 89Zr-Rituximab PET Scans"

_diagnostics, 2022, doi:10.3390/diagnostics12030596_

Round 1

Reviewer 1 Report

The authors present a paper that explores the use of convolutional neural networks to denoise PET scan with a lot of noise (low dose F18-FDG and Zr-89 rituximab) PET scans. The findings showed better signal to noise ratio compared to Gaussian smoothening which is traditionally used to denoise PET images or bilateral filtering which provides better denoising but is difficult to optimize in a generic fashion.

The authors give a good background  describe their methods and present tables to support their discussion. Their conclusions are within the finding from their findings.

A few points to note:

1.This was a study from 2 centers but in the description in the text under methods and table one demographic features of patient from Amsterdam UMC (20 FDG and 4 Zr-89 rituximab) only were described. Could the authors provide the demographic details of all patients LPC in Belgium.

2. The meaning of the (14) in line 77 not clear could you please clarify

Author Response

The authors would like to thank the reviewer for the constructive comments that helped improve the quality of the manuscript.

Point 1: This was a study from 2 centers but in the description in the text under methods and table one demographic features of patient from Amsterdam UMC (20 FDG and 4 Zr-89 rituximab) only were described. Could the authors provide the demographic details of all patients LPC in Belgium.

Response 1: We thank the reviewer for pointing this out. We have added this information to the manuscript.

Point 2: The meaning of the (14) in line 77 not clear could you please clarify.

Response 2: The PET data from LPC was prospectively obtained, and therefore, a reference is included in the manuscript to the original article. However, we did not use the square brackets. We edited this accordingly.

Reviewer 2 Report

I think it is very good research. However, there were a few parts that were difficult to read and understand, so I pointed them out. Please refer to the attached file.

Author Response

The authors would like to thank the reviewer for the constructive comments that helped improve the quality of the manuscript.

Point 1: Introduction: This chapter provides sufficient information about the subject of your research. However, for Lines 68-70, there are previous studies on human subjects, so the description needs to be changed. In addition to CNN based denoising, papers that use terms such as deep learning reconstruction would be relevant (Ann Nucl Med. 2022 Jan 14). There are many other studies that have been applied to actual cases that are not presented in this literature.

Response 1: We thank the reviewer for pointing this out. The reviewer is right that there are other studies that have applied deep learning based denoising/reconstruction algorithms on human subjects. Therefore, we now have mentioned relevant literature, including the one you recommended, and we edited Line 68-70 in the manuscript

Point 2: Participants: I think it is important to know at least how the FDG uptake were in the included case, and the distribution and number of lesions. It is unclear whether only the main lesions or all lesions were used in the later analysis, and although NSCLC from LPC was used, the references in this article (No. 14) indicate Stage 1-4. I am wondering if Stage 1 includes tumors with low accumulation or small lesions such as adenocarcinoma in situ or invasive mucinous adenocarcinoma. Rectal cancer is also a concern, whether it is an advanced cancer or has metastasis. The high accumulation of bladder in particular is an important factor in this study as it is a noise-sensitive site. I wonder if the malignant lymphoma is DLBCL, and if indolent lymphoma is included.

Response 2: For the 18F-FDG PET scans, only the primary lung tumour was delineated for analysis. For the 89Zr-rituximab PET scans, five patients with diffuse large B cell lymphoma (DLBCL) non-Hodgkin lymphoma were included. We added this in the manuscript. We also added a Figure (Figure A1) in which the distribution of the tumour characteristics (SUVBW and tumour volume) is plotted for both the 18F-FDG as well as the 89Zr-rituximab PET scans.

Point 3: Participants: The notation of cc suddenly appears in the text, but why don't you include the size, number or distribution of lesions in the table? I thought it would be better to show axial images of some representative cases. Also, you use data from LPC, but there is no one who belongs to LPC, is that okay?

Response 3: We have now defined cc (cubic centimeter) in the text. We agree that the volume of the tumours can best be presented in the table (Table 1). As mentioned in the previous section, a Figure in Appendix A (Figure A1) is also added for tumour (volume) distribution visualisation.

We also agree with the reviewer that by adding axial images, the interpretation and visualisation of the denoising results will improve for the reader. Therefore, we now present axial images from a representative case (Figure 4) in the manuscript, and moved the old Figure 4 to Appendix A (Figure A2).

Limburg PET-Center Hasselt Belgium (LPC) is part of Ziekenhuis Oost Limburg. Prof. dr. Liesbet Mesotten was responsible for the data from Ziekenhuis Oost Limburg, and is stated in the authors list.

Point 4: Model architecture: I think Line 130-131 is not well explained. Since this is a fundamental part of the study, please explain it more clearly with figures or cite past studies that used the same method.

Response 4: We thank the reviewer for pointing this out. Our network is based on the architecture of a traditional U-net model, but we adapted/modified the down sampling method and the kernel size. To support this, we added literature, and additional information to the manuscript. In addition, in Appendix B we added the Python script (Table B1) of the architecture of our model. We hope this will make it more clear for the reviewer and readers to reproduce our model architecture.

Point 5: Model performance: Line 150-153, I don't understand it immediately when it's in percentages. Please indicate the actual number or write it together.

Response 5: In addition to the percentages, the actual values or numbers are now also presented.

Point 6: Line 166 tumor recovery (TR) means recovery coefficient?

Response 6: We agree with the reviewer that it should be tumour recovery coefficient (TRC). We both changed this in the text as well as in the figures.

Point 7: Line 174-176 ACCURATE tool: This is mentioned in the JNM meeting report by one of the co-authors, but I have no way of knowing the details of it because it is not available to the public or sold as software. Is it possible to measure the VOI automatically or do I have to specify each lesion manually? VOI measurement is one of the interests of many researchers, and I think it would be better to add some more explanations or figures. Supplemental data would be fine.

Response 7: We thank the reviewer for addressing this. The ACCURATE tool is open-access which is available on sharing platform Zenodo including instruction videos. We added the reference to the Zenodo share of ACCURATE in the manuscript.

Point 8: Results, Discussion, Conclusions: well explained. However, I think it would be better to emphasize a little more that ANN-SLC and ANN-LC are clinically as good as FC. I thought that simply stating that CNN is superior to BF and GS in terms of reconstruction methods would have little clinical impact despite the length of the article.

Response 8: We agree that the LC-CNN denoised 18F-FDG PET scans are almost clinically as good as the FC 18F-FDG PET scans. This is even more the case for the SLC-CNN denoised 89Zr-rituximab PET scans, for which a higher clinical performance was seen than for the FC 89Zr-rituximab PET scans. We tried to better emphasize this in the text.

Point 9: Figure 1: The figure is illustrative, however some more explanation is needed in the main text.

Response 9: We added some additional information related to the model architecture in the main text and provided the actual python code for the model in Appendix B (Table B1).

Point 10: Figure 2, 3: It's small, cluttered, and hard to see.

Response 10: We increased the size of the figures, and scaled the Y-axis (TRC), making it less clustered and easier to visualize.

Point 11: Figure 4: Even though this is a study examining the accumulation of lesions, I think it is inappropriate as a figure because the lesion is small, and the visual impact is on the liver uptake. GS images should be kept for visual impact, but don't two BF images, FWMH4mm and 5mm are not necessary? Since it is the core of this study, it would be better to make the image more visually appealing.

Response 11: We agree that the focus of the figure is not as much directed to the tumour, and therefore, we changed it to an axial orientation and provided a better visualization of the tumour.

Point 12: Line 92 Amsterdam: Amsterdam UMC

Response 12: We changed it to Amsterdam UMC.

Point 13: Line 115 EANM: This is an abbreviation, so please spell out the official name the first time it appears.

Response 13: We changed the abbreviation to the full name.

Reviewer 3 Report

This study aimed to develop, train, and evaluate the performance of convoluted neural networks (CNS) in de-noising clinical low dose FDG and Z-89-rituximab PET/CT scans. The authors presented sufficient justification to develop this technique. The authors reported better performance for the novel CNN algorithm compared to the traditional Gaussian smoothing and bilateral filtering. The study investigated a subject that has clinical significance and may be of interest to the broader readership of this journal.

Specific comments

  1. Introduction: “Convolutional Neural Networks (CNN) are a specialized type of Neural Networks that use convolution to extract features from the PET scan.” This definition of CNN appears inadequate especially for readers who may not be familiar with this filed. I suggest the authors provide a more comprehensive definition of CNN to give potential readers a useful background insight to the technique evaluated in this study.
  2. Institution review board statement: please include the information about approval numbers and date of approval for the IRB approvals gotten for the different cohorts of patients included in this study. These are requirements by this journal.
  3. Table 1: This table only shows the demographic characteristics of the patients recruited from Amsterdam UMC. Please include similar information for the patients recruited from Belgium as well.
  4. Figure 2: Please provide large versions of the individual charts included in this figure. It is difficult to make out the information contained in them due to their small sizes.
  5. Figure 3: Same concern as in comment 4 above regarding figure 2.

Author Response

The authors would like to thank the reviewer for the constructive comments that helped improve the quality of the manuscript.

Point 1: Introduction: “Convolutional Neural Networks (CNN) are a specialized type of Neural Networks that use convolution to extract features from the PET scan.” This definition of CNN appears inadequate especially for readers who may not be familiar with this filed. I suggest the authors provide a more comprehensive definition of CNN to give potential readers a useful background insight to the technique evaluated in this study.

Response 1: We thank the reviewer for pointing this out. We edited this section to provide a more comprehensive definition of CNN in this manuscript.

Point 2: Institution review board statement: please include the information about approval numbers and date of approval for the IRB approvals gotten for the different cohorts of patients included in this study. These are requirements by this journal.

Response 2: We added the IRB approvals for all three studies.

Point 3: Table 1: This table only shows the demographic characteristics of the patients recruited from Amsterdam UMC. Please include similar information for the patients recruited from Belgium as well.

Response 3: We added this in the table.

Point 4: Figure 2: Please provide large versions of the individual charts included in this figure. It is difficult to make out the information contained in them due to their small sizes. Figure 3: Same concern as in comment 4 above regarding figure 2.

Response 4: We agree with the reviewer that the figures are difficult to read. Therefore, we increased the size of the figures, and scaled the Y-axis (TRC). We hope this will make it less clustered and easier to visualize.